# Projected impacts of climate change on the range and phenology of three culturally-important shrub species

**Janet S. Prevéy** [1,2]*, **Lauren E. Parker** [3], **Constance A. Harrington** [2]

**1** U.S. Geological Survey, Fort Collins, Colorado, United States of America, **2** Pacific Northwest Research Station, USDA-Forest Service, Olympia, Washington, United States of America, **3** USDA California Climate Hub, John Muir Institute of the Environment, University of California Davis, Davis, California, United States of America

* jprevey@usgs.gov

**Data Availability Statement:** All data files are available from: Prevey, J.P., Parker, L.E., and Harrington, C.A. 2020, Database of location and phenology data for beaked hazelnut (Corylus cornuta), Oregon grape (Mahonia aquifolium), and

## Abstract

Climate change is shifting both the habitat suitability and the timing of critical biological events, such as flowering and fruiting, for plant species across the globe. Here, we ask how both the distribution and phenology of three food-producing shrubs native to northwestern North America might shift as the climate changes. To address this question, we compared gridded climate data with species location data to identify climate variables that best predicted the current bioclimatic niches of beaked hazelnut (*Corylus cornuta)*, Oregon grape (*Mahonia aquifolium*), and salal (*Gaultheria shallon*). We also developed thermal-sum models for the timing of flowering and fruit ripening for these species. We then used multi-model ensemble future climate projections to estimate how species range and phenology may change under future conditions. Modelling efforts showed extreme minimum temperature, climate moisture deficit, and mean summer precipitation were predictive of climatic suitability across all three species. Future bioclimatic niche models project substantial reductions in habitat suitability across the lower elevation and southern portions of the species' current ranges by the end of the 21st century. Thermal-sum phenology models for these species indicate that flowering and the ripening of fruits and nuts will advance an average of 25 days by the mid-21st century, and 36 days by the late-21st century under a high emissions scenario (RCP 8.5). Future changes in the climatic niche and phenology of these important food-producing species may alter trophic relationships, with cascading impacts on regional ecosystems.

## Introduction

Understory shrub species are an integral component of terrestrial forest communities, because they contribute to biodiversity, increase carbon sequestration [1], provide food for animals [2], and create structure for nesting and shelter [3,4]. Many species of shrubs are also utilized by people around the world for food, medicines, and other important cultural and economic reasons [5–8]. However, despite the importance of food-producing understory shrub species,

salal (Gaultheria shallon): U.S. Geological Survey data release, https://doi.org/10.5066/P9G0UTKF.

**Funding:** JSP and CAH received a grant through the U.S. Geological Survey, Northwest Climate Science Center, grant number: USGS G17PG00111, USFS Agreement 17-IA-11261993-099 website: https://www.usgs.gov/land-resources/climate-adaptation-science-centers/northwest-casc. CAH received a grant through the Yakama Nation, grant number: USFS Agreement 17-CO-11261993-100, website: http://www.yakamanation-nsn.gov/. The funders had no role in study design, data collection and analysis, decision to publish, or preparation of the manuscript.

**Competing interests:** The authors have declared that no competing interests exist.

limited research has focused on how climate change may affect the distribution and phenology of these species in the future [9]. Food-producing shrubs species are vital for cultural uses by many Indigenous Peoples, and food for pollinators, frugivores, and other animals, and can serve a critical role in ecological relationships [10,11]. Consequently, understanding how climate change may alter where these shrubs grow and when they produce flowers and fruit is foundational to understanding both the possible cascading impacts on ecosystems, as well as to preparing Indigenous communities for potential disruptions to their cultural practices.

The distribution and phenology of temperate plant species has already changed markedly over the recent past in response to climate change [12–14]. Moreover, multiple studies of primarily tree species have predicted continued future shifts in distribution and phenology in the future as the climate continues to change [15–19]. However, to date, few predictive species distribution or phenology models have been made for understory shrub species. One issue that has limited examination of climate impacts on food-producing shrub species is a lack of data on where these species currently grow, and observations of when they are in flower and fruit. Emerging online databases and participatory scientist initiatives, however, now allow for the collation of many observations to be used to predict current general ranges, model future range shifts, and develop phenology models to predict current and future phenology.

Here, we used a wealth of publicly-available location, phenology, and climate data to analyze climate change effects on the potential distribution and phenology of three shrub species native to the Pacific Northwest of North America: beaked hazelnut (*Corylus cornuta)*, Oregon grape (*Mahonia aquifolium*), and salal (*Gaultheria shallon*). We selected these three species because they are common across the region, have different flowering and fruiting phenologies, and were identified as important food-producing species in a survey of tribes in Washington, Oregon, northern California, Idaho and western Montana. We recently completed a related research effort on climate effects on black huckleberry (*Vaccinium membranaceum*; 9).

The phenologies of these species differ markedly: beaked hazelnut produces catkins in very early spring, but ripe hazelnuts are not produced until late summer or fall; Oregon grape flowering and fruiting occurs relatively early, with April flowers and June-July ripe fruits; and salal flowering and fruiting occurs from mid-summer through fall. Identifying how climate change may differentially impact shrub species with differing life histories will show the potential range of responses within plant communities more broadly, and provide insights on how the cultural uses and ecosystem services they provide may be altered in the future.

## Materials and methods

### Species data sources

We collected and analyzed location and phenology data from multiple climate, forest inventory, herbaria, and participatory science sources to develop climate-envelope models and phenological models for beaked hazelnut, Oregon grape, and salal following the methods of [9]. We collected presence and phenology observations for the species from long-term data from federal agencies, the Global Information Biodiversity Facility (GBIF; [20–22]), the Consortium of Pacific Northwest Herbaria [23], the USA National Phenology Network [24], the Wilbur L. Bluhm Plant Phenology Study [25], and iNaturalist [26], among other sources (Table 1).

### Climate envelope models

To create current and future projected habitat distribution models for each species, we followed the methods of [9] and used a presence-only maximum entropy approach (MaxEnt, version 3.3.3; [32]). MaxEnt uses geo-located species presence data and environmental data to quantify the species' environmental niche, which is defined as the probability of species

**Table 1. Data sources for occurrence and phenology observations for hazelnut, Oregon grape, and salal, and past and future modelled climate data used in this study.** The species columns indicate the total number of records obtained from each data source for each species. Following the number of occurrence records, the number of records used in the climate-envelope models after spatial-filtering at a 5-km radius is listed in parentheses. Full citations for all publicly available data sources are listed in the References.

| Data type | Data source | Hazelnut | Oregon Grape | Salal | Date range |
|---|---|---|---|---|---|
| Occurrence | BLM Continuous Vegetation Survey [27] | 488 (89) | N/A | 1331 (142) | 1997–2011 |
| Occurrence | GBIF [20–22] | 1544 (280) | 972 (309) | 1413 (316) | 1838–2008 |
| Occurrence | US Forest Service R-6 Ecology Program [28] | 4411 (802) | N/A | 8363 (716) | 1979–2013 |
| Occurrence | Forest Inventory and Analysis [29] | 1379 (251) | 218 (35) | 3121 (380) | 2003–2015 |
| Occurrence | Consortium of PNW Herbaria [23] | 325 (59) | 507 (323) | 483 (208) | 1890–2015 |
| Occurrence | US National Park Service | 58 (10) | N/A | 463 (47) | 1999–2016 |
| Phenology | Wilbur L. Bluhm Study [25] | 15 | 60 | 37 | 1960–2016 |
| Phenology | US Forest Service R-6 Ecology Program [28] | N/A | N/A | 351 | 1980–2013 |
| Phenology | iNaturalist [26] | 10 | N/A | N/A | 2003–2017 |
| Phenology | Consortium of PNW Herbaria [23] | 35 | 93 | 35 | 1980–2015 |
| Phenology | Olympia Forestry Sciences Laboratory | 17 | N/A | 13 | 2015–2017 |
| Phenology | USA National Phenology Network [24] | 26 | N/A | 99 | 2010–2015 |
| Climate | AdaptWest Project [30] | N/A | N/A | N/A | 2020s, 2050s, 2080s |
| Climate | Daymet [31] | N/A | N/A | N/A | 1980–2017 |

occurrence. We utilized MaxEnt for species distribution modelling of our shrub species because it has been shown to produce accurate results using presence-only data [33,34]. MaxEnt assumes a random sample of species occurrences across the geographic domain of the study area, which we define by the domain of the gridded bioclimatic variables. However, because of the spatial clustering of species location data, environmental bias can be introduced, possibly reducing the model's predictive ability (e.g. [35,36]). To reduce this possible bias, we spatially filtered location data by 5-km radii.

To model the bioclimatic niche of each species, we acquired 30-year climatological averages of 16 bioclimatic variables from AdaptWest over the western United States and Canada (spatial extent: 32˚ to 62˚ N, 146˚ to 105˚ W;[30,37,38]. These data are interpolated on a 1-km grid, where the contemporary climate is based on the Parameter-Elevation Regressions on Independent Slopes Model (PRISM; [39], and future climate is based on downscaled global climate model (GCM) data from the Coupled Model Intercomparison Project phase 5 (CMIP5; [40]). Our analyses were conducted over the contemporary climatological period (1981–2010), the mid-21st century period (2041–2070), and the late-21st century period (2071–2100). Analyses over future periods were performed using ensemble data from 15 climate models, and were limited to experiments run under Representative Concentration Pathway 8.5. We chose RCP 8.5 as emissions trajectories to date have more closely followed RCP 8.5 than the lower RCP options [41].

Following work by [42] and [43], MaxEnt was run with our suite of 16 bioclimatic predictor variables (Table 2), performing a jackknife test of variable importance to identify the least contributing predictor, which was then removed. The predictor with the least contribution was defined as the variable with the lowest decrease in the average training gain when omitted. This stepwise process was continued until only one predictor was used for the model. For each model, the 95% confidence interval of the training gain was calculated to identify difference between models [42]. The model with the most parsimonious number of predictors that resulted in an AUC > 0.8 and whose 95% confidence interval overlapped with that of the full model was selected to create the final distribution model for each species [42,44]. For modelling the future distribution of each species, the model trained on the contemporary (1981–

**Table 2. The 16 bioclimatic variables used in analysis of habitat suitability.** Variables used in the final climate enve-
lope models for hazelnut, Oregon grape, and salal are followed by superscripts [h], [g], and [s], respectively.

| Bioclimatic Variables |
| --- |
| Annual heat moisture index (AHM) [h,g] |
| Degree-days below 0˚C (DD0) |
| Degree-days below 5˚C (DD5) |
| 30-year extreme minimum temperature, ˚C (EMT) [h,g,s] |
| 30-year extreme maximum temperature, ˚C (EXT) [g] |
| Frost-free period, days (FFP) [h,s] |
| Hargreave's climatic moisture index (CMD) [h,g,s] |
| Hargreave's reference evapotranspiration (EREF) [s] |
| Temperature differential, ˚C (TD) |
| Mean annual precipitation, mm (MAP) [g,s] |
| Mean annual temperature, ˚C (MAT) [g] |
| Mean summer (May–September) precipitation, mm (MSP) [h,g,s] |
| Mean temperature of the coldest month, ˚C (MCMT) |
| Mean temperature of the warmest month, ˚C (MWMT) [g] |
| Precipitation as snow, mm (PAS) |
| Summer heat moisture index (SHM) |

2010) climate variables was "projected" into future climate space by applying that model to the
gridded future climate data.

Following [33], we utilized most of MaxEnt's default settings, which are detailed in Merow
et al. (2013). However, we restricted model features to linear and quadratic features in order to
produce models that are easier to interpret, and that may provide a better reflection of the rela-
tionships between climate and plant species distribution [45, 46]. Using a random subsample
of 25%, we completed 10 replicated runs for each model; the subsampling approach was
selected after experiments showed it to perform better (based on AUC) than *k*-fold cross-vali-
dation. For visualization and analyzing geographic changes in species climate niche, we used
the 10-replicate median model output. To quantify habitat suitability, we selected the cumula-
tive output option in MaxEnt for analysis as it avoids the assumptions made by the logistic out-
put and is more easily interpreted than the raw output [45]. The cumulative output provides
the relative suitability of species presence at a site and ranges from 0 (low relative suitability) to
100 (high relative suitability); however, we note that the computation of the cumulative output
from the raw output means that large variations in cumulative value do not necessarily equate
to large variations in the relative probability of presence [33].

## Phenology models

To predict how dates of flowering and fruiting may change in the future, we used the collected
phenology observations and interpolated daily mean temperature data from Daymet [31] to
identify the mean accumulated daily temperature sums above a base temperature of 0 ˚C, or
'thermal sums' (also referred to as growing-degree days), from January 1ˢᵗ until the date of
flowering or fruiting for each species and phenological event. Thermal sum models assume
that plant development advances as thermal units, or growing-degree days, accumulate, and
they are widely applied in horticultural [47,48] and ecological studies [49]. We followed the
methods of [9] and accessed 1 km x 1 km daily interpolated climate data from 1980–2017 from
Daymet [31] using the phenoR package [50] in the statistical analysis program R [51]. We then

calculated thermal sums as the sum of daily mean temperatures extracted from the 1 km x 1 km Daymet grid cells above 0 ˚C from January 1st through the day of each phenology observation from 1980–2017, and used the mean value of all observations for each species and each phenological event to get thermal sum models using the TT function in the phenoR package. Thermal-sum models were used because our phenology observations were heterogeneous over time and space, and were not collected in a systematic matter at the same sites over time. We chose not to test more complex process-based models with these observational data as they may give misleading results [52]. However, even though the observations used here do not necessarily indicate the first or peak observation of a phenological event, we argue that calculating the mean accumulated thermal sums across the many observations can give an informative look at the actual accumulated temperatures required prior to peak flowering or peak fruiting [53]. We created a map showing the day of year (DOY) on which the thermal sum thresholds for each species and phenological event were met in the year 2000 (as a proxy for the "current" time period: 1981–2010) using historical gridded climate data across western North America. We then clipped the maps of estimated phenological dates for the current time period to each species' current distribution to show projections for the current timing of flowering and fruiting, and to compare to future projected phenological dates.

Next, we used the thermal sum models for flowering and fruiting to predict how climate change may alter phenology in the future using projected daily temperatures for the mid-century (2055) and end-of-century (2085) periods, for the RCP 8.5 emissions scenario, from the NASA Earth Exchange global daily downscaled projections, following the methods of [50] and [9]. We calculated thermal sum models that used each of 15 CMIP5 daily downscaled projections to arrive at CMIP5 model-mean projected flowering and fruiting dates for each future time period, as well as the standard error in projections across the 15 models. We then calculated changes in phenological events based on the difference between current and future projected phenological dates. Finally, we created maps showing the projected difference in flowering and fruiting dates by mid-century and the end of the century and clipped the extent of each future phenology map to the future-projected species distribution to illustrate how both species distribution and phenology may change in the future.

We also looked at the trends in phenology observations for the three species from a long-term dataset from the Wilbur Bluhm Plant Phenology Study in Salem, Oregon. Over the past 50 years, Wilbur Bluhm recorded the timing of many phenological events of plants around Salem, Oregon [54]. Here, we looked at this study's records for flowering and fruiting for our species of interest that spanned at least 10 years to see how phenological changes in the recent past compare to our projections for the future. Specifically, we looked at how the dates of first flowering of salal (1998–2016), first flowering for Oregon grape (1963–2016), ripe fruits of Oregon grape (2000–2016), and ripe fruits for salal (2000–2016) have changed over time. We examined change over time for each species and phenological event separately with year as the predictor variable and the date of the phenological event as the response variable using linear models in R [51]. All maps of climate envelope and phenology model results were created using ArcGIS® ArcMap 10.7.1 software (Esri, 2019).

## Results

### Species distribution

The best-fit species distribution models for the current time period assigned high relative measures of habitat suitability to areas with current occurrences of each species, and also had relatively high measures of model fit, with an AUC of 0.91 for beaked hazelnut, and 0.92 for Oregon grape and salal (Fig 1). The best-fit model for beaked hazelnut selected 5 variables as

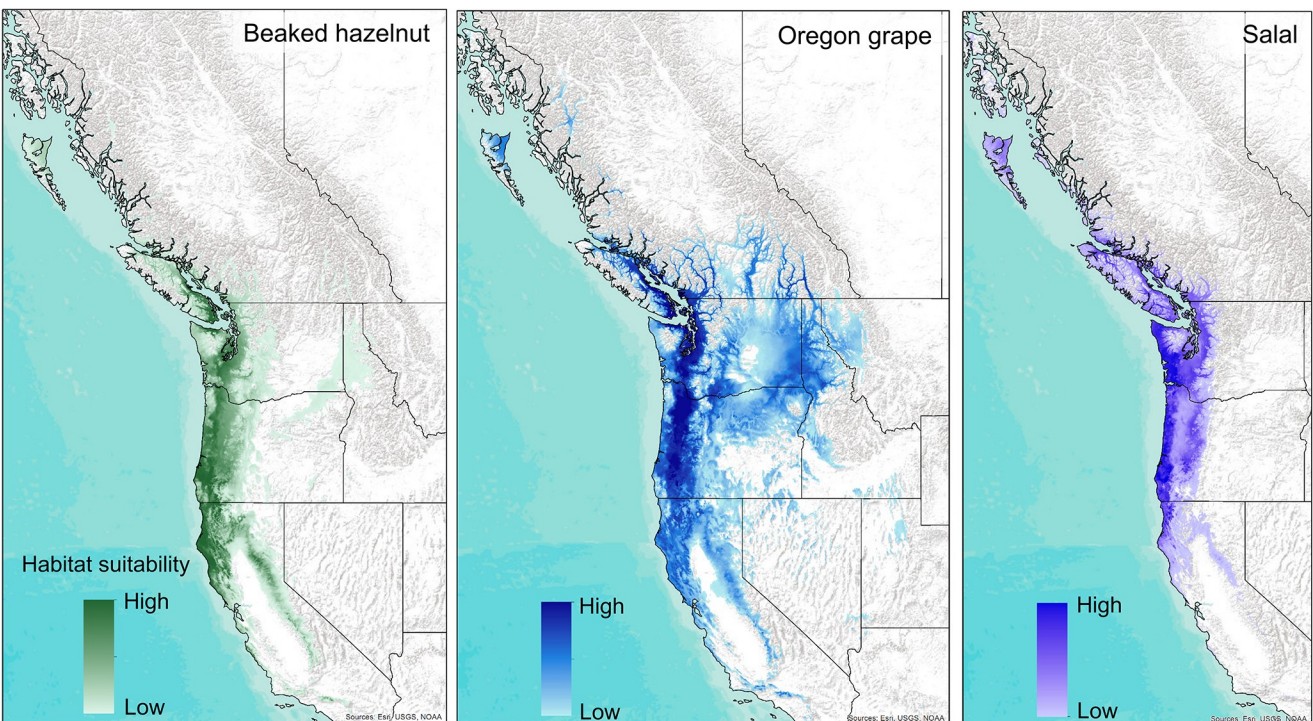

**Fig 1. Habitat suitability models across western North America for the recent time period (1981–2010) for beaked hazelnut, Oregon grape, and salal.**
Background map used: World Terrain Base; data sources: Esri, USGS, NOAA; Republished under a CC BY license with permission from ESRI original copyright [2009].

important predictors of habitat suitability: annual heat-moisture index, extreme maximum temperature over 30 years, frost-free period, Hargreaves climatic moisture deficit, and mean summer precipitation, (Table 2, S1 Fig). The best fit model for Oregon grape selected 8 variables as important predictors of habitat suitability: annual heat moisture index, extreme minimum temperature over 30 years, frost-free period, Hargreaves climatic moisture deficit, mean annual precipitation, mean summer precipitation, mean coldest month temperature, and mean warmest month temperature (Table 2, S2 Fig). The best fit model for salal selected 5 variables as important predictors of habitat suitability: extreme minimum temperature over 30 years, frost-free period, Hargreaves climatic moisture deficit, Hargreaves reference evaporation, and mean summer precipitation (Table 2, S3 Fig).

The projections for future species distributions for all three species showed a reduction in habitat suitability in coastal regions, at more southerly latitudes, and at lower altitudes, and an expansion of suitable habitat in more northerly latitudes and at higher elevations by mid-century and continuing through the end of the century (Fig 2). For beaked hazelnut, habitat suitability is projected to decrease at lower elevations and latitudes by up to 50% by the end of the century and increase at higher elevations and inland northern latitudes by up to 30% under the RCP 8.5 scenario (Fig 2). For salal, habitat is projected to decrease by 10–60% and increase by up to 50% in the Oregon and Washington Cascades and higher altitudes of Vancouver Island by the end of the century, however, the region of increasing habitat suitability for salal is smaller than for the other two species (Fig 2). For Oregon grape, habitat is projected to decrease at lower elevations and latitudes by up to 40% and increase by up to 50% at higher altitudes and latitudes by end of the century (Fig 2). For all three species, results for the mid-

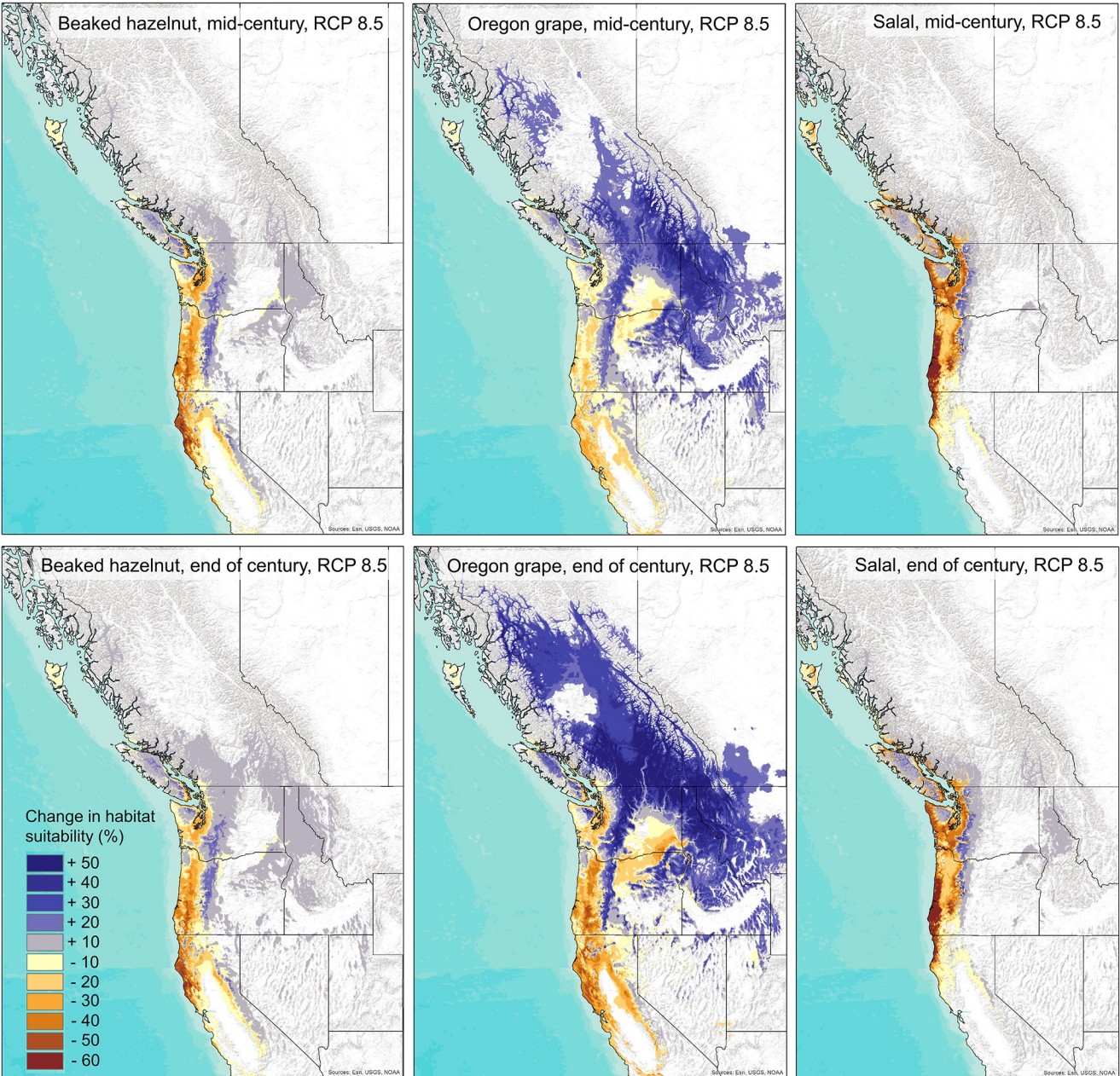

**Fig 2. Projected change in habitat suitability across western North America under the RCP 8.5 emissions scenario for beaked hazelnut, Oregon grape, and salal by the mid-21st century (top panels), and by the end of the 21st century (bottom panels).** Background map used: World Terrain Base; data sources: Esri, USGS, NOAA; Republished under a CC BY license with permission from ESRI original copyright [2009].

century time period were similar, but the models projected less reduction in habitat in warmer locations and less expansion of habitat in colder locations by mid-century than the end of the century (Fig 2).

## Phenology

Thermal-sum models projected almost a two month range in the timing of phenological events for all shrubs in the recent past across their current ranges, with earlier dates of flowering and

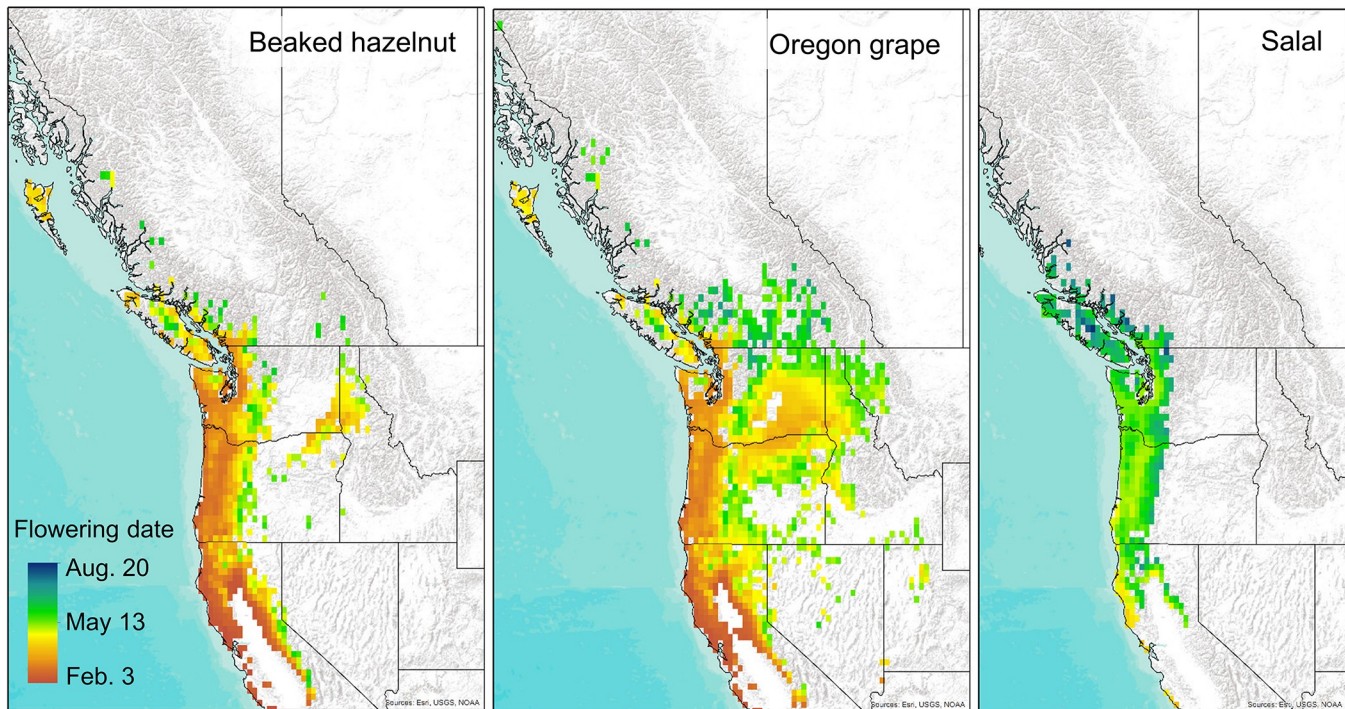

**Fig 3. Projected dates of flowering across western North America for the recent time period (1980–2010) for current ranges of beaked hazelnut, Oregon grape, and salal.** Background map used: World Terrain Base; data sources: Esri, USGS, NOAA; Republished under a CC BY license with permission from ESRI original copyright [2009].

fruiting projected for southern, coastal, and lower altitude regions and later dates in higher altitudes and more northern regions (Figs 3 and 4, S1 Table). For the recent time period (1980–2010), thermal sum models projected flowering dates ranging from early February through early May, and fruiting dates from late June through late August across the current range of beaked-hazelnut; flowering dates from mid-May through late July and fruiting dates from late June through August for salal, and flowering dates from February through June, and fruiting dates from May through August for Oregon grape (Figs 3 and 4).

Thermal-sum models projected mean advances in flowering for all species by 18 days (± 4 days) by mid-century and 35 days (± 5 days) by the end of the century under the RCP 8.5 emissions scenario (Fig 5, S4 Fig). Fruiting was projected to advance by 20 days (± 5 days) by mid-century and 37 days (± 5 days) by the end of the century (Fig 6, S5 Fig). Advances in flowering dates for species varied by ca. 43 days, and fruiting dates by ca. 32 days across projected future ranges, with greater advances in phenology predicted for inland and higher altitude locations.

Trends in phenological events from the Wilbur L. Bluhm Plant Phenology Study differed for each species. The strongest trend was for flowering of Oregon grape, which has advanced by an average of 52 days from 1963 to 2016 (slope = -1.07 ± 0.13, $F_{1,38}$ = 66.27, $P$ < 0.0001, Fig 7). Fruiting of Oregon grape has also significantly advanced over the past 16 years (slope = -1.38 ± 0.61, $F_{1,13}$ = 66.27, $P$ = 0.04, Fig 7). However, there were no significant trends in the timing of flowering and fruiting of salal over the shorter period they were recorded ($F_{1,15}$ = 1.92, $P$ = 0.186, and $F_{1,8}$ = 1.92, P = 0.78, respectively, Fig 7).

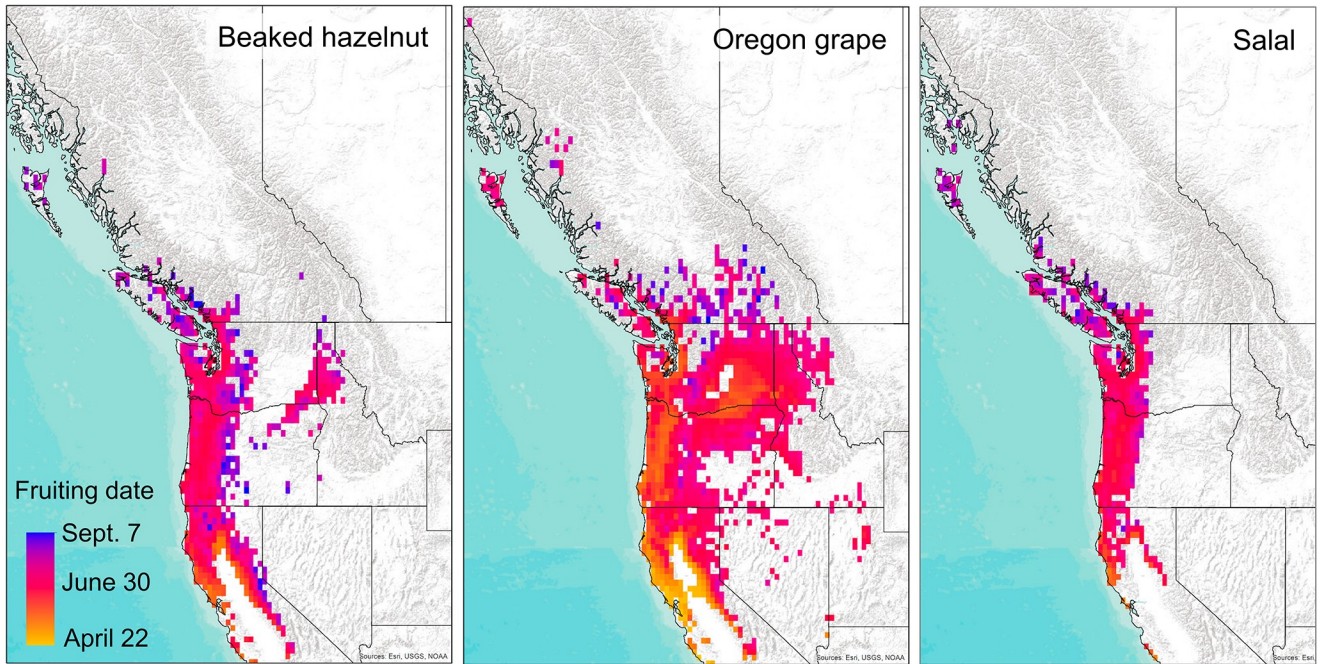

**Fig 4. Projected dates of fruiting across western North America for the recent time period (1980–2010) for current ranges of beaked hazelnut, Oregon grape, and salal.** Background map used: World Terrain Base; data sources: Esri, USGS, NOAA; Republished under a CC BY license with permission from ESRI original copyright [2009].

## Discussion

### Species distribution

Our climate envelope models predicted that hazelnut, Oregon grape, and salal will experience reductions in the potential suitability of their current habitats, and to some extent increases in habitat suitability in more northern and higher-altitude regions. The current suitable habitats for each species are overlapping, yet distinct, with hazelnut predicted to occur in mostly coastal regions extending down to California, salal predicted to occur in the smallest current region along the coast of mainly Oregon and Washington, and Oregon grape predicted to have a larger current area of suitable habitat across much of the mid-elevation regions of the Pacific Northwest. Future climate projections show the least gain in suitable habitat for salal, in addition to declining habitat suitability through much of its current range. Although salal is a very common and abundant understory species in coastal Northwestern forests, changing climate conditions could lead to a decline in abundance, with numerous impacts on the ecosystems where it is currently abundant. Besides serving as a food source for people and wildlife, salal is also extensively harvested as a part of the floral trade [5,8], and management concerning the future sustainability of salal harvesting should consider future loss in habitat with climate change. On the other hand, Oregon grape has a much broader current suitable habitat than salal as it currently grows in many different locations across elevational gradients. Our models project that Oregon grape has the greatest potential to expand its range in the future, if factors such as dispersal, population establishment, and competition with other species are not limiting. Climate envelope models for huckleberry, another shrub species with a broad range, similarly showed potential for expansion to more northern regions [9]. These results highlight the

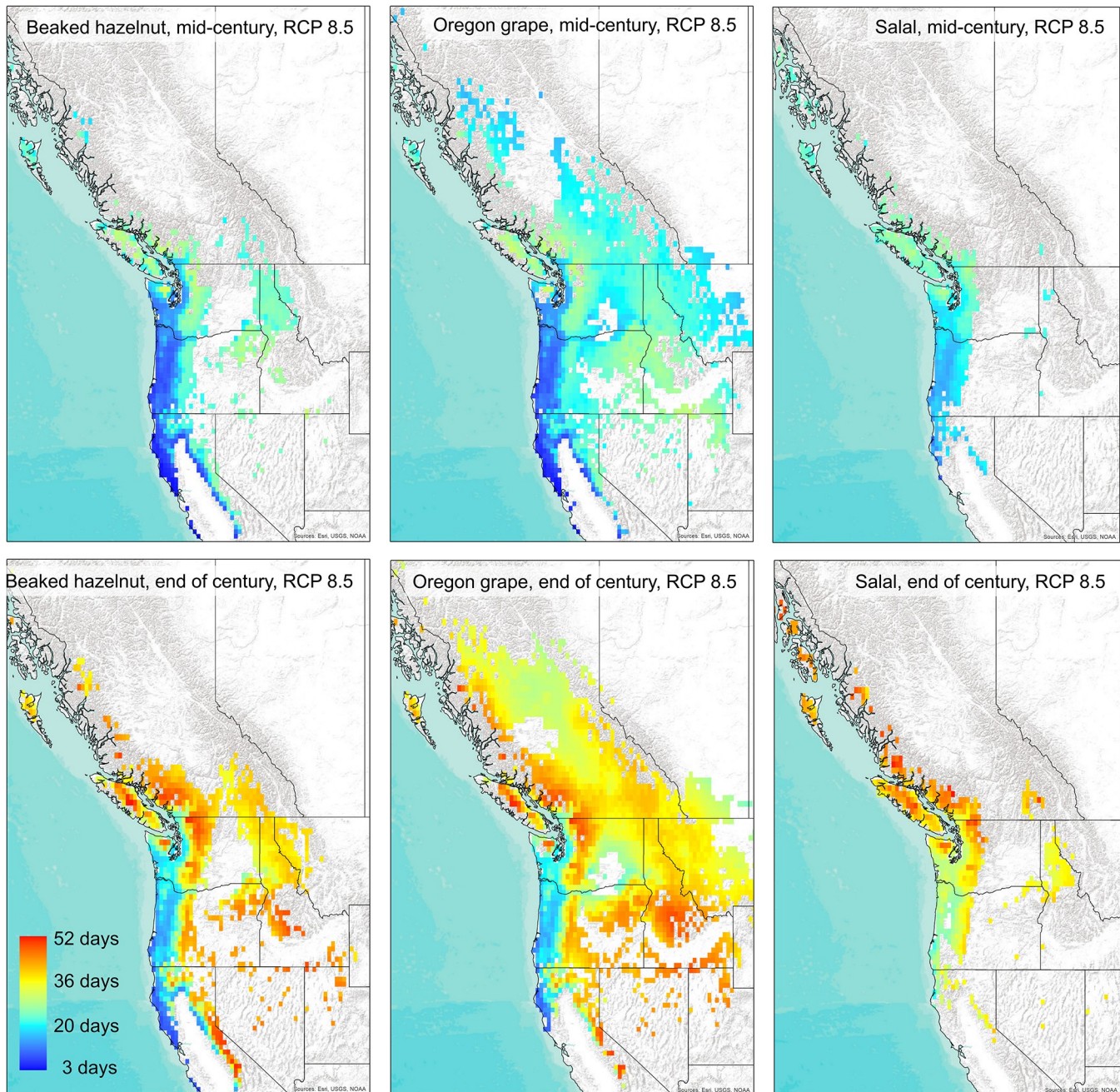

**Fig 5. Projected advance in flowering dates in future projected habitat across western North America under the RCP 8.5 emissions scenario for beaked hazelnut, Oregon grape, and salal by the mid-21st century (top panels), and by the end of the 21st century (bottom panels).** Background map used: World Terrain Base; data sources: Esri, USGS, NOAA; Republished under a CC BY license with permission from ESRI original copyright [2009].

differential impacts climate change may have on species with more specialized, versus more generalized, habitat requirements [55,56].

For all three species, extreme minimum temperatures, climatic moisture deficit, and mean summer precipitation were important variables for their final habitat suitability models. These results further confirm the importance of both damaging cold temperatures and drought as

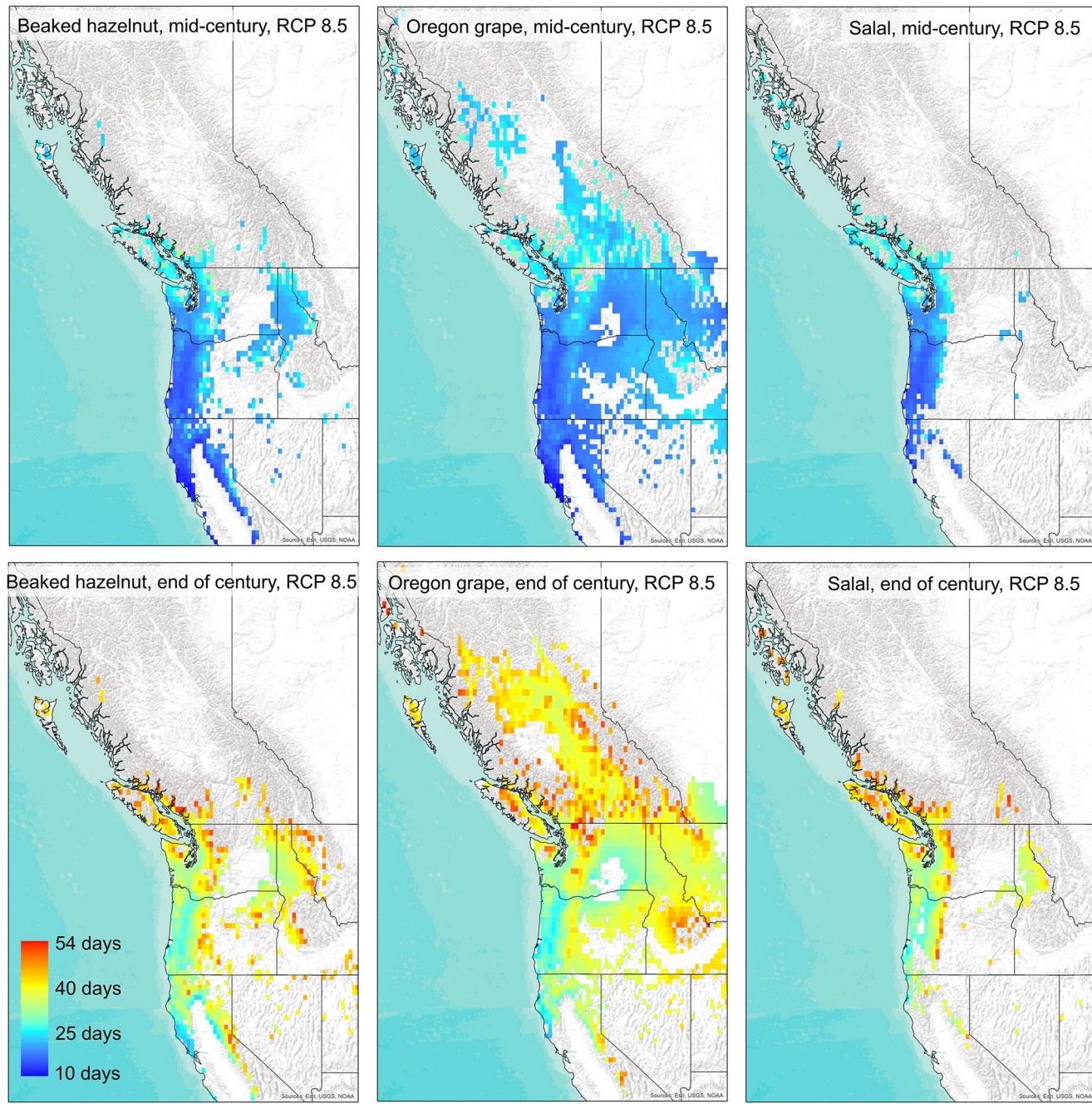

**Fig 6. Projected advance in fruiting dates in future projected habitat across western North America under the RCP 8.5 emissions scenario for beaked hazelnut, Oregon grape, and salal by the mid-21st century (top panels), and by the end of the 21st century (bottom panels).** Background map used: World Terrain Base; data sources: Esri, USGS, NOAA; Republished under a CC BY license with permission from ESRI original copyright [2009].

critical in defining the range of plant species [57,58]. As temperatures will generally increase year-round with climate change, this could influence frost risk, and increase the number of regions prone to drought, which in turn will have a large impact on the distribution of many plant species. These increases in winter temperatures and drought risk may lead to more extreme range shifts than just poleward advances in the future [59].

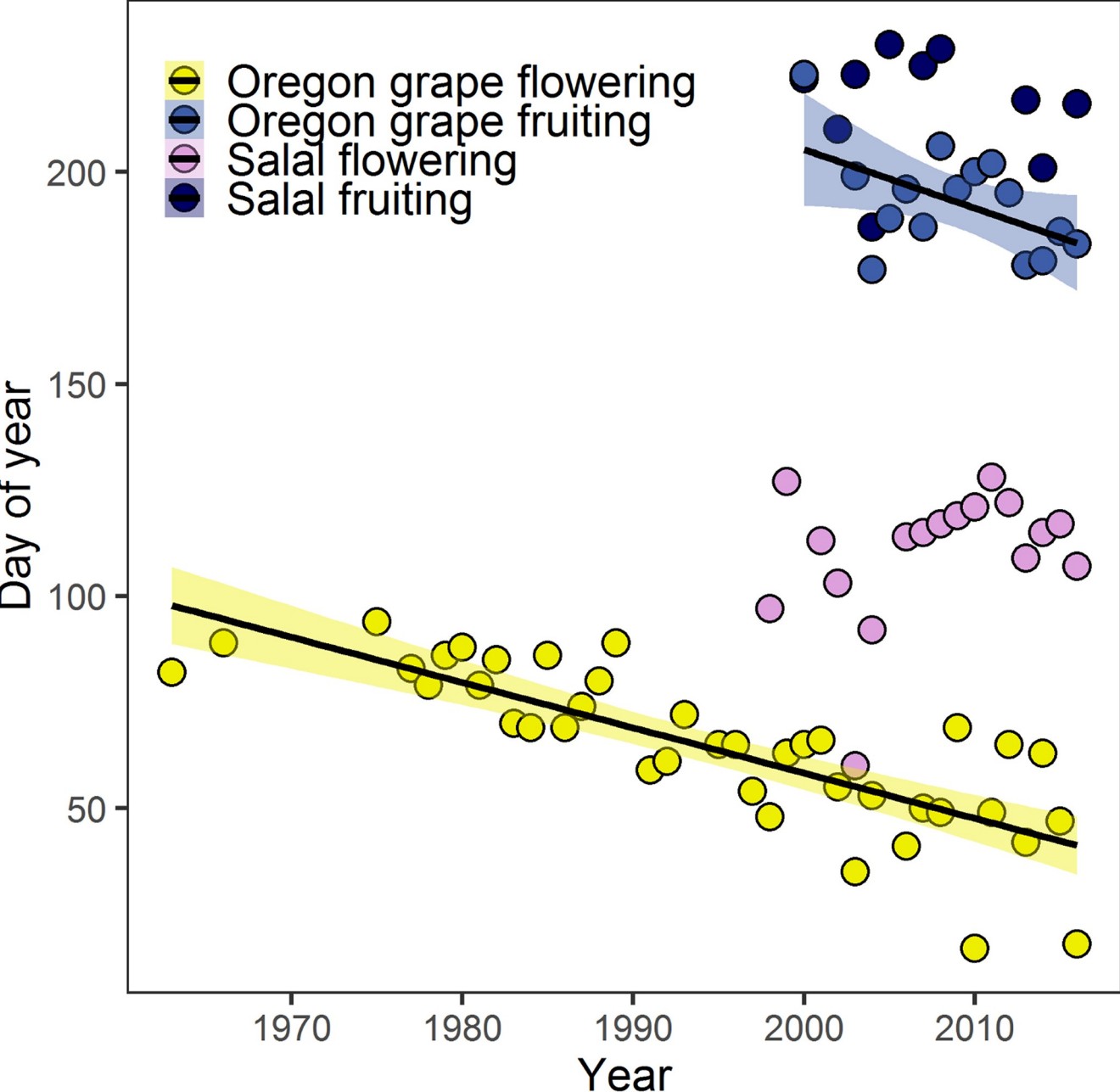

**Fig 7. Observed dates of flowering and fruiting of Oregon grape and salal from the Wilbur L. Bluhm Plant Phenology Study.** Lines represent slopes and shading indicates 95% confidence intervals for correlations significant at $P < 0.05$.

## Phenology

Results from the phenology models show strikingly large potential advances in both flowering and fruiting of the shrub species by the end of the 21st century. These advances may alter the synchrony of flower availability for pollinators, or the timing of food availability for herbivores and human harvesters. Plant species that coexist in ecosystems have evolved to stagger the timing of flowering and fruiting ripening, and the length of times fruits are available, to maximize

resource use and the chances of pollination and seed dispersal by animals [60]. In turn, birds and other animals have adapted to move across the landscape to use different food sources at different times throughout the year to maximize energy efficiency and gain [11]. Changes to the current flowering and fruit-ripening times may alter these time-dependent species interactions [61]. Our thermal-sum models show that with increasing accumulations of warm temperatures over the course of the summer, the later-ripening fruits of salal may ripen earlier (Fig 6), and shorten the window of fruit availability between the earliest fruiting (Oregon grape) and latest fruiting species (salal), which could shorten the time-window of food availability for dispersers across the landscape.

As a function of reduced snowpack, lower albedo, and increased solar absorption, higher elevations are more likely to experience increased warming with climate change than lower elevations at comparable latitudes, with greater amplification of warming under RCP 8.5 [62]. Accordingly, our projections show more rapid accumulations of thermal sums for flowering and fruiting at higher elevations as compared to lower elevations. In addition, the phenology of plants in colder, higher altitude ecosystems may be more sensitive to temperature changes than those at lower, warmer elevations [63]. These differential changes in phenology across temperature gradients have consequences for an increase in homogeneity in the timing of open flowers for pollinators and fruit resources across elevational gradients. Further, reduced winter snowpack can lead to lower soil moisture later in spring and summer [64], which could potentially influence berry productivity [65]. Collectively, these changes may impact the timing of harvesting by humans, or how animals move across the landscape to obtain food [11].

Climate is one of the primary drivers of plant growth, survival, and phenology. Warmer temperatures have already altered the phenology of plant species around the world [66,12,67]. In Salem, Oregon, where temperatures have been increasing steadily since the 1950s [54], Oregon grape has also shifted first flowering dates by more than 10 days per decade. These observations from the Wilbur Bluhm Phenology Study show that large shifts in phenological timing for at least one species have occurred in the Pacific Northwest over the recent past, and provide support for our future projections. However, these shifts are not evident for all species [54]. There were no significant trends in phenological timing for salal in the observations from Salem, Oregon, although the period of observations for salal was 18 years, much shorter than the 53 years of observations for Oregon grape. The varied periods of record for Wilbur Bluhm Phenology Study shows the importance of taking long-term measurements on ecological phenomena, as patterns and trends in these very heterogeneous data may only emerge after many years of data collection [13,54].

## Caveats

As with all modelling efforts, these projections are subject to a number of constraints and sources of uncertainty, and it is important to understand the limitations of such models for their effective use in land management planning [68]. For example, although the cumulative output from the MaxEnt modelling approach is effective for identifying shifts in overall range and for providing qualitative estimates of increases or decreases in site suitability, the output's quantitative, absolute measure of change in site suitability should be interpreted with care. Further, our species distribution models do not consider how competition between species, future land use change, soils, or disturbance may impact changing ranges in the future. In addition, incorporating differences in adaptive responses for different genotypes of the species across their current ranges could be useful for creating more realistic predictions for how they may compete and survive in areas deemed lower quality habitat in the future. There are also multiple limitations to the scope of our phenology models, as there are currently few repeated

measurements of phenological events over multiple years for these species. Although warm temperatures are often the dominant factor affecting the timing of phenology of temperate plants, the amount of cold temperatures experienced over winter (chilling), daylength (photo-period), water availability, and extreme weather events may also play a role in the timing of spring and summer phenological events of many species [60,69]. Our predictions may be inaccurate if warmer temperatures in the future also reduce the amount of winter chilling and lead to delays in flowering for shrub species with high chilling requirements. Additionally, photo-period may limit advances in phenology such that our model predictions of change will be greater than actual phenological shifts, particularly for shrub species that are more sensitive to photoperiod cues. As more precise phenology measurements are taken, phenology models incorporating chilling and photoperiod requirements can be developed for these species as well. However, despite the limitations of our modelling approaches, the models shown here are an important step towards spatial predictions of how both the locations and timing of phenology of species may change in the future, and our results provide important landscape-scale information for management and planning of restoration and other projects.

## Importance

To date, climate impact studies of culturally-important food-producing plant species have been limited, and this has hampered our ability to predict future changes in these resources and plan for effective management and restoration. The habitat suitability models developed here, based on location data from multiple publicly-available sources, provide a much more detailed estimation of the current ranges of these species than what was available prior to our work (e.g. [70]. This work is also important because it provides spatial projections for both habitat suitability and phenological changes across the entire ranges of these three phenologically diverse species. This enables comparisons between how both habitat suitability and the timing of flowering and fruiting may by impacted by climate change in the future, and allows for a more complete picture of climate impacts on important plant species across large spatial scales. For example, our climate-envelope models show that habitat suitability is projected to increase at higher elevations and latitudes, in the same places where our phenology models show the greatest potential advances in flowering and fruiting. Large advances in flowering, particularly at high altitudes and latitudes, could greatly increase the risk of frost damage and loss of fruit crops. [71].

## Applications

Results of our project will help resource managers and Indigenous Peoples understand how climate change is likely to affect the potential locations and timing of fruit production of culturally-important shrub species. Successful restoration projects should consider how climate change may impact success of desirable species [72], and our models can be used to identify regions where climate change might significantly affect the growth, survival, and timing of fruiting of food-producing shrubs, and help identify regions where these species are likely to thrive in the future. This research is also directly applicable to climate impact statements that are currently being developed by many Tribes, First Nations, National Forests, National Parks, and other entities. Our models provide an important new source of information on how climate might impact important understory species that have often been overlooked in previous climate assessments, and we hope that our research will spur more efforts to monitor and model habitat and phenology changes of other shrub species.

## Supporting information

**S1 Table. Mean thermal sums (sum of daily mean temperatures above 0 ˚C), and earliest and latest observations of flowering and fruiting of beaked hazelnut, Oregon grape, and salal.** (DOCX)

**S1 Fig. Relationships between predicted habitat suitability and the 5 climatic variables in the best-fit species distribution model of hazelnut: (A) annual heat-moisture index, (B) Hargreaves climatic moisture deficit (C) extreme minimum temperature, (D) frost-free period, and (E) mean summer precipitation.** These plots reflect the dependence of predicted suitability both on the selected variable and on dependencies induced by correlations between the selected variable and other variables. (DOCX)

**S2 Fig. Relationships between predicted habitat suitability and the 8 climatic variables in the best-fit species distribution model of Oregon grape: (A) annual heat moisture index, (B) Hargreaves climatic moisture deficit, (C) extreme minimum temperature, (D) frost-free period, (E) mean annual precipitation, (F) mean summer precipitation, (G) mean coldest month temperature, and (H) mean warmest month temperature.** These plots reflect the dependence of predicted suitability both on the selected variable and on dependencies induced by correlations between the selected variable and other variables. (DOCX)

**S3 Fig. Relationships between predicted habitat suitability and the 5 climatic variables in the best-fit species distribution model salal: (A) Hargreaves climatic moisture deficit, (B) extreme minimum temperature, (C) Hargreaves reference evaporation, (D) frost-free period, and (E) mean summer precipitation (Table 2, S4 Fig).** These plots reflect the dependence of predicted suitability both on the selected variable and on dependencies induced by correlations between the selected variable and other variables. (DOCX)

**S4 Fig. Standard error in projected flowering dates between 15 climate models across western North America under the RCP 8.5 emissions scenario for beaked hazelnut, Oregon grape, and salal by the mid-21[st] century (top panels), and by the end of the 21[st] century (bottom panels).** Background map used: World Terrain Base; data sources: Esri, USGS, NOAA; Republished under a CC BY license with permission from ESRI original copyright [2009]. (DOCX)

**S5 Fig. Standard error in projected fruiting dates between 15 climate models across western North America under the RCP 8.5 emissions scenario for beaked hazelnut, Oregon grape, and salal by the mid-21[st] century (top panels), and by the end of the 21[st] century (bottom panels).** Background map used: World Terrain Base; data sources: Esri, USGS, NOAA; Republished under a CC BY license with permission from ESRI original copyright [2009]. (DOCX)

**S1 Data.** (PDF)

## Acknowledgments

We thank Leslie Brodie, Yianna Bekris, Jacob Strunk, and Beverly Luke for assisting with gathering, editing, and analyzing presence and phenology data. We thank Dominique Bachelet and

Nikolas Stevenson Molnar for identifying possible datasets, and Koen Hufkens for assistance with calculating thermal sum models. We thank the following sources: USFS Forest Inventory and Analysis program, USFS R-6 Ecology Program, US National Park Service, USDI Bureau of Land Management, USA National Phenology Network, the Consortium of PNW Herbaria, and Wilbur L. Bluhm Plant Phenology Study for data on plant occurrences and phenology. Any use of trade, firm, or product names is for descriptive purposes only and does not imply endorsement by the U.S. Government.

## Author Contributions

**Conceptualization:** Janet S. Prevéy, Constance A. Harrington.

**Data curation:** Janet S. Prevéy.

**Formal analysis:** Janet S. Prevéy, Lauren E. Parker.

**Funding acquisition:** Janet S. Prevéy, Constance A. Harrington.

**Investigation:** Janet S. Prevéy.

**Methodology:** Janet S. Prevéy, Lauren E. Parker.

**Resources:** Constance A. Harrington.

**Validation:** Lauren E. Parker.

**Writing – original draft:** Janet S. Prevéy, Lauren E. Parker, Constance A. Harrington.

**Writing – review & editing:** Janet S. Prevéy, Lauren E. Parker, Constance A. Harrington.

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
