## [Decision Letter · Decision Letter 0]

9 Jan 2020

PONE-D-19-32397

Projected impacts of climate change on the range and phenology of three culturally-important shrub species

PLOS ONE

Dear Dr. Prevey,

Thank you for submitting your manuscript to PLOS ONE. After careful consideration, we feel that it has merit but does not fully meet PLOS ONE’s publication criteria as it currently stands. Therefore, we invite you to submit a revised version of the manuscript that addresses the points raised during the review process.

The two reviewers, expert in the field, raise a number of interesting and consistent suggestions to make stronger, more complete and convincing the manuscript. Please consider them carefully and explain if and how you applied them into the new version.

We would appreciate receiving your revised manuscript by Feb 23 2020 11:59PM. To enhance the reproducibility of your results, we recommend that if applicable you deposit your laboratory protocols in protocols.io, where a protocol can be assigned its own identifier (DOI) such that it can be cited independently in the future. For instructions see: http://journals.plos.org/plosone/s/submission-guidelines#loc-laboratory-protocols

We look forward to receiving your revised manuscript.

Kind regards,

Sergio Rossi

Academic Editor

PLOS ONE

Journal Requirements:

3. Please upload a new copy of Figure 3 as the detail is not clear. Please follow the link for more information: http://blogs.PLOS.org/everyone/2011/05/10/how-to-check-your-manuscript-image-quality-in-editorial-manager/

4. Please upload a copy of Figure 7, to which you refer in your text. If the figure is no longer to be included as part of the submission please remove all reference to it within the text.

5. We note that Figures 1-6 in your submission contain map images which may be copyrighted.

a.    You may seek permission from the original copyright holder of Figures 1-6 to publish the content specifically under the CC BY 4.0 license. 

Reviewers' comments:

Reviewer's Responses to Questions

**Comments to the Author**

1. Is the manuscript technically sound, and do the data support the conclusions?

Reviewer #1: Partly

Reviewer #2: Yes

2. Has the statistical analysis been performed appropriately and rigorously? 

Reviewer #1: Yes

Reviewer #2: Yes

3. Have the authors made all data underlying the findings in their manuscript fully available?

Reviewer #1: Yes

Reviewer #2: No

4. Is the manuscript presented in an intelligible fashion and written in standard English?

Reviewer #1: Yes

Reviewer #2: Yes

5. Review Comments to the Author

Reviewer #1: It is important to understand habitat suitability, distribution, phenological mechanism and responses to climate change for the species with social and economic values, to inform assessment of future impact and management planning. This study applied MaxEnt and Thermal-sum models to estimate habitat suitability and flowering and fruiting phenology of three shrub species, which provided interesting results. More information should be stated to prove the reasoning of applied methods. More evidence and discussion should be provided to prove the importance and how useful the research findings could be in this study.

There are different phenology models for spring phenology including different sets of variables. This manuscript needs more arguments to state the reason why the authors selected thermal-sum model for flowering and fruiting phenology of the three shrub species. How much do we know about the phenological mechanism of the shrub species so far? Are there any evidence supporting the thermal-sum model is the best for fitting flowering and fruiting phenology? If we know little of phenological mechanism, how can we use the future predictions based on the models with uncertain response mechanism?

The information of phenology data processing should be clear. How were all different phenology observation records collected in multiple sites incorporated into spatial grid cells? Were thermal-sum models fitted at grid cell level or site level or using the whole data?

In comparisons of current and future phenology, which values were used? Yearly estimation was made, then calculated the mean date? E.g. mean phenological dates during 1981 and 2010 representing the current period? Or used one specific year estimation in comparisons?

Line 374-379. Argument is weak. Relate to the comment above. Need to state the reason why this simple phenology model was used and how the limitations may affect the results and conclusions. While the model used growing-degree-day as the only driver of phenology and warming trend was the only driver of future change, the advanced results were straightforward. But is it realistic that flowering and fruiting time will be advanced without constrains over time? What important message should we take from the current results?

Line 23-24. Phenology means timing of biological events. Should be “the timing of critical biological events” or “phenology”

Line 126. What are 16 bioclimatic variables? Need to refer to Table 2 here.

Line 236-238. What do 2055 and 2085 time period mean? Are they yearly estimation in the two specific years or mean estimation for the whole time periods 2041-2070 and 2071-2100?

Line 279-285. Could not find Figure 7.

Line 302. Need to explain how the results of least gain in habitat suitability suggested a decline of abundance of salal.

Line 308. Should specify “other factors”, such as population establishment and competition with other species.

Line 328-331. Should clarify the point from this sentence. Which results suggested more advanced fruiting of latest fruiting species than earliest fruiting species, and then shorten the time period between the earliest and latest fruiting.

Line 352-354. Remind the specific periods of record for Oregon grape and salal here, so the readers would know how exactly short the observations record was.

Line 372-373. Water availability and extreme weather events (e.g. drought and heat) may also affect phenology, which was rarely included in prior phenology models.

Reviewer #2: In this report, the authors use basic species distribution models and simple phenology models to extrapolate the current distribution and phenology of three culturally important shrubs species in a warmer future. While the study is generally clearly written and the methods are presented concisely, I feel that it lacks some novelty, as, to a large part, it seems to be quite similar to recently published study on huckleberry (AFM 2109, by the same authors. Referenced in this manuscript) just with different species three other different species. As such, the question remains if the authors exhaustively searched for all available data or, if even there is still more data on further species available, e.g. to cover all shrub species of the western US. It would also be interesting ton include the results from the previous study into the current analysis and focus on the differences between the four species.

Apart from that, my main critique is, that the authors seem to analyze and discuss the species distribution and phenology separately. I would like to see a deeper discussion of how a shifted phenology might interact with the bioclimatic parameters selected for the species distribution modelling in the long run, especially when making prediction for the end of the century. Are the currently selected bioclimatic parameters still valid in a future climate with strongly shifted phenology? For example, are late winter freezing spells becoming a more of a problem when the phenology is shifted much earlier? Furthermore, the author claims that due to obvious limitations of the chosen very simple models (for which they provide a good rationale), this is just a first step to predict the responses into the future. With the authors being aware of the limitations, it would be important to discuss the most likely responses with more realistic models and their consequences for their predictions as well, also given that the direction in which the additional factors may influence the responses can be estimated.

All in all, it is a brief, solid study, but needs a major revision of the discussion, which in turn, could also lead to a much stronger conclusion.

Some minor details:

The dataset is not available yet, doi missing.

There are a few wrongly formatted references in the introduction

6. PLOS authors have the option to publish the peer review history of their article (what does this mean?). If published, this will include your full peer review and any attached files.

Reviewer #1: No

Reviewer #2: No

---

## [Author Response · Author response to Decision Letter 0]

30 Mar 2020

Dear Sergio Rossi and reviewers,

Thank you for your thoughtful edits and suggestions. We’ve addressed each point made by the editor and reviewers below, with our responses in bold italic. We’ve also highlighted all changes made to the manuscript in red in the ‘Revised Manuscript with Track Changes’ file. 

Sincerely,

Janet Prevey, Lauren Parker, Connie Harrington 

Editor and reviewer comments:

Journal Requirements:

 The manuscript meets the style requirements.

We will add the DOI for the data as soon as the data is approved through the USGS data release review process, and prior to publication of the manuscript. No changes are needed for the data availability statement.

All data files are available from: Prevey, J.P., Parker, L.E., and Harrington, C.A. 2020, Database of location and phenology data for beaked hazelnut (Corylus cornuta), Oregon grape (Mahonia aquifolium), and salal (Gaultheria shallon): U.S. Geological Survey data release, https://doi.org/XX.XXXX/XXXXX.

3. Please upload a new copy of Figure 3 as the detail is not clear. Please follow the link for more information: http://blogs.PLOS.org/everyone/2011/05/10/how-to-check-your-manuscript-image-quality-in-editorial-manager/

We’ve uploaded higher quality versions of all Figures, including Fig. 3.

4. Please upload a copy of Figure 7, to which you refer in your text. If the figure is no longer to be included as part of the submission please remove all reference to it within the text.

 We’ve uploaded Fig. 7.

5. We note that Figures 1-6 in your submission contain map images which may be copyrighted. 

All map images were created by the authors in ESRI’s ArcGIS 10.1; no copyrighted material was used. One can use static maps that include ArcGIS Online maps hosted by Esri in academic publications without the need for written permissions: https://doc.arcgis.com/en/arcgis-online/reference/static-maps.htm

The one caveat is that the authors must provide attribution on or near the map or image that includes an ArcGIS Online map hosted by Esri. We have included this attribution in each panel of each Figure that uses an ESRI background map, and added this statement to the end of the Methods (l. 200-201): “All maps of climate envelope and phenology model results were created using ArcGIS® ArcMap 10.7.1 software (Esri, 2019).”

5. Review Comments to the Author

Reviewer #1: It is important to understand habitat suitability, distribution, phenological mechanism and responses to climate change for the species with social and economic values, to inform assessment of future impact and management planning. This study applied MaxEnt and Thermal-sum models to estimate habitat suitability and flowering and fruiting phenology of three shrub species, which provided interesting results. More information should be stated to prove the reasoning of applied methods. 

We have added additional information to the Methods section explaining the reasoning for the applied methods we chose for species distribution modeling here: l. 105-106, and reasoning for why we chose thermal sum models here: l. 167-171).

More evidence and discussion should be provided to prove the importance and how useful the research findings could be in this study.

This research is directly applicable to climate impact statements being developed by many Tribes of the Northwest, as well as National Forests, National Parks, and other regional climate impact assessments. Our models provide an important new source of information on how climate might impact important understory species that have often been overlooked in previous climate assessments. We’ve added several specific examples for why these results are important and how these results can be used to the end of the discussion (l. 408-416).

There are different phenology models for spring phenology including different sets of variables. This manuscript needs more arguments to state the reason why the authors selected thermal-sum model for flowering and fruiting phenology of the three shrub species. 

Thermal-sum models were used because our phenology observations were heterogeneous over time and space, and were not collected in a systematic matter at the same sites over time, thus it would be difficult to test a suite of models that include other, more complex, cues such as chilling and photoperiod. We were hesitant to test more complicated process-based models with these observational data as they may give misleading results (Hänninen et al., 2019). Additionally, thermal-sum, or growing degree day, models have been used extensively for horticultural and ecological studies, and have accurately predicted the flowering and fruiting dates of at least two shrub species (White et al., 2012; Laskin et al., 2019). While we are aware of the plethora of phenological models that incorporate multiple cues, such as chilling temperatures and photoperiod, we do not have any evidence that those cues are more important than forcing temperatures, and thus decided to use simple models with temperature sums. We’ve added this additional justification to the methods, (l. 167-171).

Citations:

Hänninen, H., Kramer, K., Tanino, K., Zhang, R., Wu, J., Fu, Y.H., 2019. Experiments Are Necessary in Process-Based Tree Phenology Modelling. Trends in Plant Science 24, 199–209. https://doi.org/10.1016/j.tplants.2018.11.006

Laskin, D.N., McDermid, G.J., Nielsen, S.E., Marshall, S.J., Roberts, D.R., Montaghi, A., 2019. Advances in phenology are conserved across scale in present and future climates. Nature Climate Change 9, 419–425. https://doi.org/10.1038/s41558-019-0454-4

White, S.N., Boyd, N.S., Van Acker, R.C., 2012. Growing Degree-day Models for Predicting Lowbush Blueberry (Vaccinium angustifolium Ait.) Ramet Emergence, Tip Dieback, and Flowering in Nova Scotia, Canada. horts 47, 1014–1021. https://doi.org/10.21273/HORTSCI.47.8.1014

How much do we know about the phenological mechanism of the shrub species so far? 

There is not much known about the phenological mechanisms for shrubs in this region. However, for most temperate plant species, warming temperatures in spring are the most important cue for the timing of phenological events. The below graph shows the relationship between mean spring temperatures and our phenology observations for the three shrub species. As you can see, for all but one species*phenological event (hazelnut flowering) – there is a strong negative relationship between spring temperatures and the day of year that the phenological events occurred; that is, the warmer the spring temperatures, the earlier in the year flowering occurs. Again, we are not trying to assume that other environmental cues are not important for phenological timing, just that we currently do not have experimental evidence to justify including them in our models. Finally, we acknowledge that the physiological mechanisms driving the development of any of these shrub species are undoubtedly a complex interplay between multiple environmental drivers. Our methods and results are one effort to identify some of these drivers and quantify their relationships to shrub phenology. 

Are there any evidence supporting the thermal-sum model is the best for fitting flowering and fruiting phenology? 

Laskin et al. 2019 used thermal sum models to predict timing of fruiting of another fruit-producing shrub in the northwestern region, with landscape-scale thermal sum models performing very well and predicting the date of fruiting and leaf senescence of Sheperdia canadensis to within 3 days. Additionally, the above graph of our observations suggests that there is a relationship between warm temperatures and earlier phenological dates. Again, there is not much known about phenological mechanisms for shrub species, and we hope that this research will encourage more detailed studies using experimental data and repeat measurements to identify other possible mechanisms.

If we know little of phenological mechanism, how can we use the future predictions based on the models with uncertain response mechanism?

Our phenology models show how flowering and fruiting may advance with warmer temperatures, and we hope they can serve as a first step for future studies seeking to identify phenological mechanisms for shrubs, and monitor phenological changes over time. We have addressed the reasoning for the soundness of thermal-sum models in the Methods (l. 165-171), and have added additional statements about the limitations of our thermal sum models in the Discussion (l. 379-384). Ultimately all models will have some level of uncertainty, but we argue that given the close observed connection between temperature and phenology across wild and cultivated species, the future predictions give a reasonable assessment of likely changes and provide resource managers and other interested stakeholders with information useful for their planning purposes. 

The information of phenology data processing should be clear. How were all different phenology observation records collected in multiple sites incorporated into spatial grid cells? 

Daily temperatures for each phenology measurement were taken from the 1 x 1 km grid from Daymet for the year of that observation. Daily temperatures from these grid cells for each phenology observation were summed from January 1st through the day of the phenological event in the year the observation was taken. We have clarified the methods to better explain the phenology data processing (l. 159-164).

Were thermal-sum models fitted at grid cell level or site level or using the whole data?

Thermal-sum models were calculated using all the data, as the sum of daily mean temperatures above 0 °C from January 1st through the day each phenology observation from 1980-2017, and we used the mean value of all observations for flowering and fruiting to get thermal sum models for each species and phenological event. We have clarified the methods to better explain the phenology data processing (l. 159-164).

In comparisons of current and future phenology, which values were used? 

In these comparisons, we compared the DOY of phenological events estimated for the year 2000 from Daymet data to phenological dates estimated for 2055 and 2085 using daily temperature data from each of 15 CMIP5 daily downscaled projections. We then calculated changes in phenological events based on the difference between current estimated DOYs and future projected DOYs of phenological events (l. 172-193).

Yearly estimation was made, then calculated the mean date? E.g. mean phenological dates during 1981 and 2010 representing the current period? Or used one specific year estimation in comparisons?

We used estimations from daily temperatures in the year 2000 to represent the current time period (l. 172-175). 

Line 374-379. Argument is weak. Relate to the comment above. Need to state the reason why this simple phenology model was used and how the limitations may affect the results and conclusions. While the model used growing-degree-day as the only driver of phenology and warming trend was the only driver of future change, the advanced results were straightforward. But is it realistic that flowering and fruiting time will be advanced without constrains over time? What important message should we take from the current results?

We now expandd upon the reasoning that the simple phenological model was used in the methods (l. 167-172), and explicitly state how the results may be inaccurate if chilling temperatures or photoperiod are stronger environmental cues for phenology in the Discussion (l. 380-384).

Line 23-24. Phenology means timing of biological events. Should be “the timing of critical biological events” or “phenology”

Changed to “critical biological events”.

Line 126. What are 16 bioclimatic variables? Need to refer to Table 2 here.

We’ve added a reference to Table 2 (l. 125).

Line 236-238. What do 2055 and 2085 time period mean? Are they yearly estimation in the two specific years or mean estimation for the whole time periods 2041-2070 and 2071-2100?

2055 and 2085 indicated a 30-year period centered on these years. That is, the mean over 2041-2070 was indicated by 2055, and 2071-2100 by 2085. However, we have now changed the wording from 2055 and 2085 to mid- and end of the century respectively to clarify this, while indicating in the first use of these terms that “mid-century” is with respect to 2041-2070, and “end of century” is with respect to 2071-2100 (l. 118-119 and l. 237-239). 

Line 279-285. Could not find Figure 7.

Apologies, we have now added Figure 7.

Line 302. Need to explain how the results of least gain in habitat suitability suggested a decline of abundance of salal.

We’ve altered the sentence to clarify: “Future climate projections show the least gain in suitable habitat for salal, in addition to declining habitat suitability through much of its current range.” L. 301-303.

Line 308. Should specify “other factors”, such as population establishment and competition with other species.

We have replace ‘other factors’ with population establishment and competition with other species (l. 312).

Line 328-331. Should clarify the point from this sentence. Which results suggested more advanced fruiting of latest fruiting species than earliest fruiting species, and then shorten the time period between the earliest and latest fruiting.

We have clarified this by referencing Fig. 6 and including the names of the species (Oregon grape and salal). “Our thermal-sum models show that with increasing accumulations of warm temperatures over the course of the summer, the later-ripening fruits of salal may ripen earlier (Fig. 6), and shorten the window of fruit availability between the earliest fruiting (Oregon grape) and latest fruiting species (salal), which could shorten the time-window of food availability for dispersers across the landscape.” l. 335-338.

Line 352-354. Remind the specific periods of record for Oregon grape and salal here, so the readers would know how exactly short the observations record was.

We now state the specific periods of record: “There were no significant trends in phenological timing for salal in the observations from Salem, Oregon, although the period of observations for salal was 18 years, much shorter than the 53 years of observations for Oregon grape.” L. 359-360.

Line 372-373. Water availability and extreme weather events (e.g. drought and heat) may also affect phenology, which was rarely included in prior phenology models.

We have added water availability and extreme weather events to the list of factors that could possibly influence phenology (l. 379).

Reviewer #2: In this report, the authors use basic species distribution models and simple phenology models to extrapolate the current distribution and phenology of three culturally important shrubs species in a warmer future. While the study is generally clearly written and the methods are presented concisely, I feel that it lacks some novelty, as, to a large part, it seems to be quite similar to recently published study on huckleberry (AFM 2109, by the same authors. Referenced in this manuscript) just with different species three other different species. 

We appreciate the reviewer’s comment and recognize the similarities between the manuscripts. However, we note that including the detailed methods (as we did for the AFM 2019 manuscript) as well as the results for all four shrub species in one publication resulted in an unwieldly, extremely long manuscript that was challenging for readers to follow. Thus, we broke up the studies to allow for more discussion of the specific results for the various shrub species. This approach of breaking large studies into multiple manuscripts is not uncommon. We also note that novelty is not a requirement of publication in PLOS One.

As such, the question remains if the authors exhaustively searched for all available data or, if even there is still more data on further species available, e.g. to cover all shrub species of the western US. 

There is an impressive amount of species location data available for Northwestern shrubs, and almost all relatively common species of plants and animals available on GBIF.org, and availability is expanding daily. However, modeling habitat suitability and phenology for all shrub species in the western US was beyond the scope of our research as we wished to focus on those species that were identified as being of significant cultural importance in the Northwest. We also want to underscore the time and computational expense involved in adding each additional species. Each species required a unique habitat suitability model to be developed using MaxEnt, and for each citizen scientist observation for each species, time was taken to view and verify that the species and phenology stage had been identified correctly before thermal sum models for species-specific phenology could be developed. So while the principal reason we constrained our analysis to three shrubs is a function of the identified cultural importance, the realities of the limitations on our time would have limited our analysis to far fewer than all shrub species across the western US even in the absence of our culturally-focused motivation for shrub selection. That said, we hope that our research will spur more efforts to monitor and model habitat and phenology changes of other shrub species. 

It would also be interesting ton include the results from the previous study into the current analysis and focus on the differences between the four species.

This is also beyond the scope of our current work, and since the previous results for huckleberry are already published, we do not want to present the results again here. However, we’ve now added comparisons to our results for habitat suitability for the current species and huckleberry here (l. 312-314) and discuss differences in phenology between the three phenologically diverse shrub species in this manuscript in the Discussion (l. 334-338).

Apart from that, my main critique is, that the authors seem to analyze and discuss the species distribution and phenology separately. I would like to see a deeper discussion of how a shifted phenology might interact with the bioclimatic parameters selected for the species distribution modelling in the long run, especially when making prediction for the end of the century. 

We now expand our discussion of how shifts in habitat suitability and phenology may interact in the Discussion (l. 400-405).

Are the currently selected bioclimatic parameters still valid in a future climate with strongly shifted phenology? For example, are late winter freezing spells becoming a more of a problem when the phenology is shifted much earlier? 

This is a good point. We now expand our discussion of how shifts in habitat suitability and phenology may interact to increase frost risk of shrub species expanding into higher altitude and latitude ecosystems (l. 4002-405).

Furthermore, the author claims that due to obvious limitations of the chosen very simple models (for which they provide a good rationale), this is just a first step to predict the responses into the future. With the authors being aware of the limitations, it would be important to discuss the most likely responses with more realistic models and their consequences for their predictions as well, also given that the direction in which the additional factors may influence the responses can be estimated.

We have added several sentences to the discussion detailing how changes in other environmental controls over phenology would alter our results. L 380-384.

All in all, it is a brief, solid study, but needs a major revision of the discussion, which in turn, could also lead to a much stronger conclusion.

We thank the reviewer for their suggestions and agree that the revisions add substantially to the discussion.

Some minor details:

The dataset is not available yet, doi missing.

We will add the DOI to the data release as soon as the data is approved through the USGS data release review process, and prior to publication of the manuscript.

There are a few wrongly formatted references in the introduction

We have corrected the references in the Introduction (l. 58).

---

## [Editor Report · Decision Letter 1]

7 Apr 2020

PONE-D-19-32397R1

Projected impacts of climate change on the range and phenology of three culturally-important shrub species

PLOS ONE

Dear Dr. Prevey,

Thank you for submitting your manuscript to PLOS ONE. After careful consideration, we feel that it has merit but does not fully meet PLOS ONE’s publication criteria as it currently stands. Therefore, we invite you to submit a revised version of the manuscript that addresses the following minor points:

Discussion could benefit to be separated in sections of specific topics identified by subtitles.

Consider to convert the last part of discussion in application. Or add a conclusion f the work.

References are too many for a research paper. Generally, 30-40 should be enough. Reduce redundant or marginal references.

Readers should understand figures without reading the legend. As much as possible, improve the figures with this aim. Some examples: Fig 1: replace A, B, C with the name of the species, quantify (add values) to the range low-high, add the location of the study area at least in the legend; Fig 2: include the names of the scenarios (and the species)

We would appreciate receiving your revised manuscript by May 22 2020 11:59PM. To enhance the reproducibility of your results, we recommend that if applicable you deposit your laboratory protocols in protocols.io, where a protocol can be assigned its own identifier (DOI) such that it can be cited independently in the future. For instructions see: http://journals.plos.org/plosone/s/submission-guidelines#loc-laboratory-protocols

We look forward to receiving your revised manuscript.

Kind regards,

Sergio Rossi

Academic Editor

PLOS ONE

---

## [Author Response · Author response to Decision Letter 1]

12 Apr 2020

Discussion could benefit to be separated in sections of specific topics identified by subtitles.

We have now separated the Discussion into topics identified by the subtitles ‘Species distribution’, ‘Phenology’, ‘Caveats’, and ‘Importance’.

Consider to convert the last part of discussion in application. Or add a conclusion f the work.

We’ve converted the last part of the Discussion into a subsection titled ‘Applications’.

References are too many for a research paper. Generally, 30-40 should be enough. Reduce redundant or marginal references.

We kindly request that we retain all the references for our manuscript. Since this research paper includes both climate envelope and phenology models, we have twice the number of necessary references to provide information for both disparate types of Methods, and many of these references were added per earlier reviewer comments. Additionally, 14 of the references are for the publicly available open-source data we used in the manuscript and are therefore necessary to include. We were not aware of any instructions to authors limiting the number of references for research papers in PLOS One. 

Readers should understand figures without reading the legend. As much as possible, improve the figures with this aim. Some examples: Fig 1: replace A, B, C with the name of the species, quantify (add values) to the range low-high, add the location of the study area at least in the legend; Fig 2: include the names of the scenarios (and the species)

We have altered the Figures to replace the letters for panels with species names and the names of the scenarios, and added the location of the studies to Figure legends. In Fig. 1, the range values are denoted as ‘low’ and ‘high’ rather than 0 and 100 because, in Maxent, the computation of the cumulative output from the raw output means that large variations in cumulative value do not necessarily equate to large variations in the relative probability of presence, and thus it is advisable to list qualitative rather than numerical data, since a value of 100% does not necessarily mean that there is 100% chance of the species occurring at that location (Phillips and Dudik 2008). We do, however, quantify numerical differences in cumulative outputs for Fig 2.

Phillips SJ, Dudík M. Modeling of species distributions with Maxent: new extensions and a comprehensive evaluation. Ecography. 2008;31: 161–175.

---

## [Editor Report · Decision Letter 2]

17 Apr 2020

Projected impacts of climate change on the range and phenology of three culturally-important shrub species

PONE-D-19-32397R2

Dear Dr. Prevey,

We are pleased to inform you that your manuscript has been judged scientifically suitable for publication and will be formally accepted for publication once it complies with all outstanding technical requirements.

With kind regards,

Sergio Rossi

Academic Editor

PLOS ONE
---

## [Editor Report · Acceptance letter]

28 Apr 2020

PONE-D-19-32397R2 

Projected impacts of climate change on the range and phenology of three culturally-important shrub species 

Dear Dr. Prevéy:

I am pleased to inform you that your manuscript has been deemed suitable for publication in PLOS ONE. Congratulations! Your manuscript is now with our production department. 

With kind regards,

on behalf of

Prof. Sergio Rossi 

Academic Editor

PLOS ONE